# Study of the Course of Cement Hydration in the Presence of Waste Metal Particles and Pozzolanic Additives

**DOI:** 10.3390/ma15082925

**Published:** 2022-04-17

**Authors:** Ina Pundienė, Jolanta Pranckevičienė, Modestas Kligys, Giedrius Girskas

**Affiliations:** Laboratory of Concrete Technology, Institute of Building Materials, Vilnius Gediminas Technical University, Linkmenų Str. 28, LT-08217 Vilnius, Lithuania; jolanta.pranckeviciene@vilniustech.lt (J.P.); modestas.kligys@vilniustech.lt (M.K.); giedrius.girskas@vilniustech.lt (G.G.)

**Keywords:** waste-metal particle, cement, pozzolanic additives, hydration, compressive strength, ultrasound propagation velocity

## Abstract

As the construction of hydrotechnical and energy facilities grows worldwide, so does the need for special heavyweight concrete. This study presents the analysis of the influence of waste-metal particle filler (WMP) on Portland cement (PC) paste and mortars with pozzolanic (microsilica and metakaolin) additives in terms of the hydration process, structure development, and physical–mechanical properties during 28 days of hardening. Results have shown that waste-metal particle fillers prolong the course of PC hydration. The addition of pozzolanic additives by 37% increased the total heat value and the ultrasound propagation velocity (UPV) in WMP-containing paste by 16%; however, in the paste with only WMP, the UPV is 4% lower than in the WMP-free paste. The density of waste-metal particle fillers in the free mortar was about two times lower than waste-metal particle fillers containing mortar. Due to the lower water absorption, the compressive strength of WMP-free mortar after 28 days of hardening achieved 42.1 MPa, which is about 14% higher than in mortar with waste-metal particle filler. The addition of pozzolanic additives decreased water absorption and increased the compressive strength of waste-metal particle filler containing mortar by 22%, compared to pozzolanic additive-free waste-metal particle fillers containing mortar. The pozzolanic additives facilitated a less porous matrix and improved the contact zone between the cement matrix and waste-metal particle fillers. The results of the study showed that pozzolanic additives can solve difficulties in local waste-metal particle fillers application in heavyweight concrete. The successful development of heavyweight concrete with waste-metal particle fillers and pozzolanic additives can significantly expand the possibility of creating special concrete using different local waste. The heavyweight concrete developed by using waste-metal particle fillers is suitable for being used in load balancing and in hydrotechnical foundations.

## 1. Introduction

According to the ACI definition [1,2], heavyweight concrete density is higher than conventional concrete due to heavyweight aggregate, which is used in heavyweight concrete components. The density of heavyweight concrete is determined by the aggregate’s specific gravity as well as the parameters of the other concrete components [3]. Concretes with specific gravities greater than 2600 kg/m^3^ and aggregates with specific gravities greater than 3000 kg/m^3^ are referred to as heavyweight concrete and heavyweight aggregate, respectively.

Heavyweight concretes with a density of 4000 kg/m^3^ and above are used for the manufacture of ballasts and weights in port construction, crane counterweight, for floating oil and gas platforms, protection elements for underwater pipelines, dams, soil stabilization, protection against electromagnetic effects, radiation, etc. [4,5,6,7,8,9]. Currently, heavy concretes used for construction works contain special costly fillers, such as barite, ilmenite, and magnetite [10,11,12]. Usually heavyweight aggregate contains a high content of metallic phases. Minerals like magnetite [13,14,15,16], hematite [17,18,19], goethite [3], limonite [3,20], colemanite [21,22], barite [23,24,25,26], lead-zinc [27,28,29,30,31], and ferrophosphorus [32] are some of the natural minerals utilized as aggregates in heavyweight concrete, whereas steel punchings and iron shot are artificial aggregates [8,33]. The density of some minerals and metallic waste is as follows: limonite is 3.4–4.0 gr/cm^3^, siderite is 4.1–4.7 gr/cm^3^, barite is 4.0–4.6 gr/cm^3^, ilmenite is 4.3–4.8 gr/cm^3^, hematite is 4.9–5.3 gr/cm^3,^ steel shot parts are 6.2–7.8 gr/cm^3^ [34,35], and according to the above research, it is suggested that metal waste materials can be used as heavyweight fillers for special concrete. 

Concrete that contains barite (BaSO_4_) is widely used in building construction, such as nuclear power stations, particle accelerators, and medical hospitals, although it is not highly feasible as there are not enough barite reserves in the world [36]. The high protective properties against the action of electromagnetic fields are obtained using special fillers with a high (35–70%) iron content, among them—barite, ilmenite, magnetite, and hematite [12]. The price of heavyweight concrete with fillers of such materials, depending on the kind of used fillers, may increase 3 ÷ 10 times compared to a common one. The heavyweight concrete prices depend on the cost of transportation and extraction of minerals and also on each country’s available raw materials. Therefore, the development of cheaper heavyweight concrete using natural resources or waste materials available in their own countries is promising for new construction materials development.

Therefore, in this study, we had to look for appropriate local waste—metal particles from punching steel sheets and stamping driving chains—suitable for use as components in heavyweight concrete. Heavyweight concrete is subject to the same product testing requirements as regular concrete, with the exception that it is considerably more likely to have quality fluctuations due to segregation. During placing, heavyweight concrete is particularly prone to segregation. Segregation of ingredients of heavyweight concrete causes differences in strength and density that are unacceptably risky for heavyweight concrete work because these variations can affect the concrete’s durability characteristics [1,19].

As a binder, different types of Portland cement (normal, pozzolanic, slag), and aluminate cement is used [30,33,37]. Another research [14] investigated the influence of the water to cement (w/c) ratio on the mechanical properties of heavyweight magnetite concrete and found that when density increases, compressive strength and modulus of elasticity improve. However, it was found [4,32,38,39,40] that an increase in heavyweight fillers reduces the strength of the material and the slump index of the concrete mix. When using heavyweight fillers, the overall strength of the concrete may decrease due to their poor adhesion of the metal to the cement [7]. The partial replacement of the conventional fillers with metallic mineral hematite fillers (at a constant water/cement ratio of 0.40) improved the slump index and compressive strength of concrete [41]. The results showed that the compressive strength increased by the increase of hematite content. Another study [42] presented research on the effects of different concentrations of hematite from 10 to 50% of volume (at the water/cement ratio of 0.42) on the physical and mechanical properties of concrete showed that hematite aggregate increased its density and durability. After 30 freeze–thaw cycles, the plain concrete lost 21.3% of its compressive strength, while the concrete containing 10% hematite lost only 7.8%. The hematite aggregate also lowers the drying shrinkage of concrete. In the study [13,43], the heavy concrete with barite aggregate was tested. A decrease in water absorption, compressive strength, and tensile strength values of concrete was determined as a result of an increase in the barite ratio in concrete composition. With the increase of barite, an increase in unit weight, modulus of elasticity, ultrasonic pulse velocity (UPV), and the coefficient of thermal conductivity of barite-based concrete was determined. Due to the increase in the ratio of barite, a decrease in Schmidt hardness, water absorption (%), compressive strength, and tensile strength values of concrete were determined [13].

Recent research has looked toward using iron waste materials (such as nano-sized magnetite, Fe_3_O_4_, superfine steel dust with high zinc oxide content, and ferritic fume dust) to substitute Portland cement in the fabrication of radiation-resistant concrete [44,45,46,47]. Minerals like bauxite, hydrous iron ore, or serpentine, which are all slightly heavier than normal-weight concrete, can be used when the high fixed water content is required [3,17].

Some research showed that an additive based on recycled steel fibres caused improved brittle behaviour of the matrix, especially in terms of toughness and post-cracking behaviour, elevated the concrete mechanical properties, particularly its dynamic and fatigue resistance, shear, and post-cracking strength [48,49,50]. Concrete with metal fillers has increased thermal resistance, better thermal conductivity, and lower shrinkage. Meanwhile, it is difficult to eliminate the shrinkage completely and prevent the appearance of cracks on the border between cement and metal fillers [3,5,12,34,36,37].

The proper disposal of metal refuse is another vital aspect of environmental preservation. Due to the high cost of manufacturing products from natural metal-containing aggregates, the use of metal waste is of increasing interest. The viability of using iron waste in heavyweight concrete has been demonstrated in studies of new construction materials with iron waste, specifically perforated steel tape, percussion caps, dust waste from filters, and metal powders [6,7,8,45]. However, the introduction of waste particles as fillers can reduce the material quality of concretes. Thus, replacing some parts of natural aggregates with steel waste particles caused some reduction in strength [11]. It was shown that as the amount of steel waste increased, the workability of the concrete mixture became an important issue which eventually required larger amounts of water to achieve a minimum slump [6,7,8]. Research showed that the 30–40% replacement of natural fillers by steel shot (density 7.8 kg/m^3^) with a diameter of 0.5 and 1 cm provided the density of concrete up to 5000 kg/m^3^; however, the compressive strength of such density samples is by 20% lower than the compressive strength of samples with density 2200 kg/m^3^ [11].

To increase the strength by improving the contact zone between the filler and cement matrix, special additives and admixtures are often used, including nano-sized ones such as nano-silica [51,52], superplasticizers [53], microsilica [7], and metakaolin [54,55,56]. The above-listed pozzolanic additives should be used in heavy concretes to improve the packing density of concrete structures, improve the physical properties of the contact zone between the filler and cement matrix, and decrease the damage of voids and cracks in the concrete [11].

The main purpose of the research is to develop and evaluate heavyweight concrete, incorporating local waste-metal particles from punching steel sheets and stamping driving chains as fillers to produce high-demand concrete from locally available materials. The process of returning waste-metal particles to the metal product manufacturing places is highly costly and inefficient. To compensate for the decrease in strength of pozzolanic additives, concrete-structure-modifying additives were employed in this research. Meanwhile, there is a lack of information on cement hydration characteristics in the presence of waste-metal particle fillers and different pozzolanic additives. The influence of pozzolanic additives on the hydration of cement in the presence of waste-metal particle fillers was tested. Such information will significantly supplement knowledge about the hardening features of this type of concrete in the early period of hydration and the factors influencing the physical and mechanical properties of heavyweight concrete.

## 2. Materials and Research Methods

In this study, waste-metal particles (WMP) from punching steel sheets and stamping driving chains from the JSC DITTON Driving Chain Factory in Daugavpils, Latvia were used as metal fillers in specimens of heavy concrete. WMP had the shape of a disk with a diameter of 3.0–5.8 mm, a height of 1.03–2.3 mm, and a density of 7.8 g/cm^3^ (Figure 1).

Portland cement CEM I 42.5 N (PC) complying with the EN 197-1:2011 requirements was used in the research. The mineral content of the PC by weight was: C_3_S—58.54%; C_2_S—15.29%; C_3_A—10.40%; Ca_4_AF—10.17%. The specific surface was 0.41 m^2^/g. The specific surface of the PC was 420 m^2^/g and the bulk density was 1.1 g/cm^3^; the setting started at 140 min and ended at 190 min, and the amount of alkali was 0.8% at maximum. The size of PC particles ranged from 1 to 100 μm and 50% of PC included 15–30 μm particles.

The additives used in the research were microsilica and metakaolin from industrial waste. Microsilica (RW-Fuller) was received from RW-Silicium GmbH, Pocking, Germany; the specific surface was 15 m^2^/g and the calcination loss at 975 °C was 0.6% (Figure 2).

Metakaolin, industrially produced by Bauchemie GmbH, Bottrop, Germany, and commercially available with a specific surface of 9.8 m^2^/g and density of 0.43 g/cm^3^, was used in the heavy concrete forming mixtures for replacing a portion of sand in the composition. The size distribution of particles (performed using a CILAS 1090) showed that about 40% of metakaolin was formed by 6–13 μm particles. The phase analysis of metakaolin, carried out by X-ray diffraction, shows that the main minerals are kaolinite and quartz (Figure 3).

The chemical compositions of the PC, microsilica, metakaolin are presented in Table 1.

Sand fractions of 0/1 and 0/4, complying with EN 12620:2003+A1:2008 requirements, were used for calorimetric and mortar compositions (Table 2). The content of fragile types of rock, such as sandstone, limestone, etc., is not higher than 2%. For mixing, the water used complied with the EN 1008:2005 requirements. Polycarboxylate-based superplasticizer (Glenium 440, BASF Corporation, Ludwigshafen, Germany) in equal amounts for all mortar compositions (1% from PC amount) was used.

For the calorimetric measurements, a differential isothermal calorimeter, the ToniCAL III, was used. Five mixes of different compositions (45 g distilled water and 100 g of solid substance, where a special sieved fraction of 0/1 mm of sand was used) were studied at 20 °C, and the heat evolution curves were registered during the hydration reaction of up to 48 h. The compositions of the five mixes are presented in Table 3. Only PC and sand were used in the composition BM-0. WMP fillers were used in the next four compositions. Part of the sand was substituted by WMP in the composition BM-1. The sand was substituted by microsilica and metakaolin additives in the composition BM-2 and BM-3 and by the microsilica and metakaolin mix (1:1) in the composition BM-4. All solid components were mixed in dry conditions before testing.

Mortar compositions were prepared in a planetary mixer. The amount of water was adjusted so that all prepared compositions had the same fluidity measured by a mini-slump cone (13 cm). Sixteen specimens (70 × 70 × 70 mm^3^ cubes) for every composition were moulded in the forms. The mould was filled in three layers, with each approximately one-third of the height of the mould and were compacted by vibration. The specimens were cured for 1 day in moulds at the temperature of (20 ± 2) °C and humidity of (95 ± 5)% and then until testing at a temperature of 20 ± 2 °C and humidity of 65 ± 5%, to avoid corrosion of WMP because it may affect the results. The components used in the study and their part of the mass are shown in Table 4. The control composition P0 without WMP fillers was prepared for comparison with the PA, PAS, PAMK, and PASMK compositions where the sand was partially replaced by WMP, WMP, and microsilica, WMP and metakaolin, and WMP with both microsilica and metakaolin, correspondingly. Prepared specimens were tested after 2, 7, and 28 days of hardening.

The ultrasound propagation velocity (UPV) was defined according to the EN 12504-4:2021 requirements. The hardening kinetics of the whole mortar mix can be continuously followed and monitored using an ultrasonic method. This technique is based on the propagation velocity of ultrasonic waves through a sample. It can be applied from the end of mixing to the completely hardened state of the mortar. The Pundit 7 ultrasonic pulse indicator apparatus and data logger were used for recording measurements. The fresh mortar mix was placed in a cylindrical mould (100 mm diameter and 40 mm high) between two ultrasonic transducers operating at 10 pulses per second and a frequency of 54 kHz. The UPV increased according to the setting of the mortar mix and the development of its hardening structure. The value of UPV in fresh mortar mix was tested for 24 h and hardened specimens were tested after 2, 7 and 28 days of hardening.

The compression strength was determined according to EN 12390-3:2019, the density according to EN 12390-7:2019, and the water absorption test was performed according to EN 12390-16:2019. The capillary water absorption coefficient (C_w_) (kg/m^2^·min) of the mortar specimens was analysed according to the standard EN 772-11:2011.

The microscopic investigations of the surface of the metal particle samples were performed with the use of a Veho Discovery DX-3 microscope, which has an optical magnification of up to 220×. To evaluate the characteristic peculiarities of adhesion of metallic particles with cement in fracture surfaces of specimens of different compositions, a SEM EVO 50 EP (resolution 1.5 nm) was used.

## 3. Results

### 3.1. Fresh Mortar Properties

#### 3.1.1. Calorimetric Investigations

Calorimetric investigations allowed for the evaluation of the influence of additives on the hydration process of PC. During the hydration process, the initial increase in heat release corresponds to the wetting period, which continuously passes to the induction stage, which is characterized by the lack of thermal effects. The sudden increase in heat release was strictly connected with exothermic massive precipitation of hydrates, followed by further hydration.

The results of the substitution of sand for WMP in the BM-1 mix compared to the BM-0 mix show that (Figure 4a) the WMP particles reduced the wetting heat release rate. The wetting heat release rate for the BM-0 mix was 2.6 W/kg, and 1.48 W/kg for the BM-1 mix. The addition of microsilica in the BM-2 mix increased the wetting heat release rate up to 1.72 W/kg, whereas the addition of metakaolin increased the wetting heat release rate just up to 1.49 W/kg. Finally, when both microsilica and metakaolin additives were used together, the wetting heat release rate reached 1.58 W/kg.

The maximum heat release rate time for the BM-0 and BM-2 mixes was reached after 10 h and 12 h. The same tendencies were observed in the research [57] when calorimetric studies of two compositions of cement and sand and cement and waste metal aggregates were performed. The presence of waste metal aggregates increases the maximum heat release rate time in cement mix by 20%, compared to the cement mix with sand. For the other BM-1, BM-3, and BM-4 mixes, the maximum heat release rate time was reached after 14 h (Figure 5a).

The replacement of sand in the BM-0 mix with WMP in the BM-1 mix resulted in a 14.5% higher maximum heat release rate value. As referred to in [58], the specific heat of the metal (0.46 KJ/kg·K) is about two times lower than for sand (0.84 KJ/kg·K). Apparently, as a result, the metal particles were quickly absorbed by the heat of hydration of the cement, and the hydration processes accelerated. When microsilica and metakaolin additives were separately used in the BM-2 and BM-3 mixes, the maximum heat release rate value was 39.1 and 23.95% higher than in the BM-0 mix.

When microsilica and metakaolin additives were used in the mix together, the maximum heat release rate reached a slightly higher (by 5%) value as in the BM-1 mix; however, it was observed that in this mixture, the maximum heat release rate time is extended up to 34 h and between 20–33 h of hydration additional heat release reaching 0.9 W/kg is observed. The same tendencies with prolonged maximum heat release rate in mixes with metakaolin and microsilica additives were observed in the research [59,60].

This indicates that the WMP fillers prolonged the PC hydration course and increased the heat release rate. When WMP fillers were used in combination with microsilica, the PC hydration course was extended by an additional 2 h, and the heat release rate accelerated up to 1.64 times. When both additives were used together, the PC hydration course was extended for an additional 2 h and the heat release rate accelerated up to 1.31 times during the 30 to 33 h. It is seen that the maximal heat release rate of PC with WMP fillers obviously accelerated in the latest hydration period when metakaolin and microsilica additives were used together.

The replacement of sand with WMP in the BM-1 mix increased the total heat by 30% after 48 h of hydration (Figure 6a). The use of WMP fillers and the microsilica additive increased the total heat value by about two times, and the same use of WMP and metakaolin additive increased the total heat value by about 1.4 times compared to the BM-0 mix. In the BM-4 mix, where microsilica and metakaolin additives were used together, the total heat value was about 1.6 times bigger compared to the BM-0 mix.

The summarized results of the hydration process are presented in Figure 4b, Figure 5b and Figure 6b. The wetting heat release rate clearly depends on the presence of WMP in the mix; WMP fillers up to 39.0% adsorb wetting heat of mixes. The maximum heat release rate value greatly depends on the WMP fillers. The presence of WMP fillers by 14.5% increased the maximum heat release rate value, the presence of WMP and microsilica additive increased the maximum heat release rate value by 59.0%, but in the mix with WMP and metakaolin, the maximum heat release rate value increased by just 17.7%. When both additives were used together, the maximum heat release rate value increased by 27.8%. It can be attributed to the alkaline medium in the presence of cement minerals. 

An increase in the alkalinity when the pH is more than 10 causes an increase in the pH on the metal (iron) surface [58]. It is possible that water first oxidizes the metal surface, and the wetting heat release rate is depressed, but furthermore, the corrosion rate decreases since iron is more and more passivated in the presence of alkalis and dissolved oxygen. The microsilica additive showed an alkaline reaction in the water solution that can more actively react with cement hydration product on the surface of the metal particles. The metakaolin additive, showing a slight acid pH in the water solution, can slightly prolong the cement hydration process. The total heat value after 48 h of hydration supports this hypothesis. It can be concluded that the WMP fillers increased the total heat value in the mixes, and both replacements of sand to waste WMP and the use of additives, especially a microsilica additive, a higher total heat value to be reached in the mixes.

#### 3.1.2. Ultrasound Propagation Velocity Method

For a complete characterization of the development of the structure of hardening concrete samples, ultrasonic tests of prepared mortars were carried out for 24 h. Researchers in [61] suggested that UPV can be used very effectively to monitor the hydration and formation of the microstructure of cement pastes. The PC hydration and hardening structure development process (by UPV method) are divided into three steps: 1—induction period, when UPV is less changing; 2—UPV is increasing sharply (quick structure compaction period); 3—UPV is slowly increasing and becoming stable.

The UPV tests of fresh mortar mixes (P0) and PA (Figure 7) show that replacing the part of sand with WMP fillers by a small amount (about 2 h) can retard the structure development during an induction period (early phase of hydration) in comparison to the control P0 composition. The UPV values for the P0 samples changed from 950 m/s to 1035 m/s during the first 1.6 h, but in the PA samples, the same changes in UPV values from 940 m/s to 1040 m/s happens during the first 3.5 h. At the end of the induction period, the UPV in P0 composition had risen sharply, but in the PA mixes, the structure development increased at a slower rate.

After 4 h of hydration, the UPV values for P0 samples reached 1445 m/s, but in the PA sample reached just 1060 m/s, which is 36.0% less than in the control P0 sample. After 24 h of hydration, the UPV values for P0 and PA samples reached 3090 m/s and 2975 m/s values. It means that WMP fillers significantly stop the structure development process during the induction period and seems that it happens because of hydration heat transferring to WMP, as can be seen from calorimetry investigations. When cement stops intensively hydrating, the metal fillers stopped to collect the hydration heat, and the structure development runs conventionally. After 24 h of hydration, the structure densification development in the sample PA reached almost the same level as in the control sample.

In the case of PAS, PAMK, and PASMK compositions, the structure formation occurred later, and the induction period was extended up to 2–2.6 h. The UPV values in the pointed compositions varied between 940–990 m/s. After 4 h of hydration, the highest UPV values of 1800 m/s were reached in the sample where both microsilica and metakaolin additives were used. In the PAMK composition, the addition of metakaolin promoted the development of the structure the least during 4 h of hydration, and the UPV values reached 1610 m/s. However, it was observed that the microsilica additive promotes the development of the structure very close to the developed PASMK sample structure, and after 4 h of hydration, the UPV values in the PAS sample reached 1745 m/s.

The beneficial influence of the microsilica additive was observed after 5–6 h of hydration when the UPV values in the PAS and PASMK samples started to sharply increase and after 24 h of hydration, reached 3420 m/s and 3400 m/s. This tendency is in agreement with calorimetry results (Figure 5a). The calorimetry research shows that the metakaolin additive had less influence on the heat release in the samples and it is visible that the metakaolin additive increased UPV values at a slower rate. After 24 h of hydration, the UPV values reached just 3060 m/s. This research indicated that the microsilica additive alone or used together with metakaolin additive increased the UPV values up to 13.0–9.6% and 7.8–4.2%, compared to the PA and control P0 sample UPV values.

These investigations showed that the WMP filler used in the mixture significantly retarded the early phase of the PC hydration process, the wetting heat release rate and the UPV values. When the WMP was used in combination with microsilica and both the microsilica and metakaolin additive, the structure development of the sample was faster, possibly due to the heat that the metal accumulates at the start of wetting and then returns to the sample, and thus accelerates the interaction between cement hydration products and the additives. The calorimetry and UPV tests confirmed such assumptions, showing a significant increase in the heat release rate, total heat, and UPV in the samples.

### 3.2. Properties of Hardened Samples

#### 3.2.1. Density and UPV

Measurements of the density of mortar samples (Figure 8) showed that, depending on the WMP and used additives, the density significantly varied. The same was observed in the study [10,12,58]. It is pointed out that with the increase in the replacement percentage of sand to the waste-iron aggregate, the density of samples gradually increased because the specific gravity of waste iron is 42.8% higher than the specific gravity of sand (12). The density of the samples was significantly affected by the density of the metal fillers and the amount used (2,3).

Replacing the part of sand with WMP fillers in the PA composition increases the density of samples by 1.9 times compared to the control sample. The additions of pozzolanic microsilica and metakaolin additives decrease the density of PAS and PAMK compositions samples, respectively, by 1.45% and 4.40%. However, in the composition PASMK, where microsilica and metakaolin mix was used, the density of samples decreased just by 0.89%. However, for the PAS, PAMK, and PASMK samples, the obtained density can also be influenced by a little different w/c value (0.55 for PA composition and 0.58 for compositions with additives) in the mortar mixture. It can also be very important to emphasize that the density results do not correlate with the obtained UPV values after 24 h of hardening. The samples with the highest density possessed the lower UPV values; this suggests that the structure of these samples was uneven and contact between WMP fillers and PC matrix was incomplete.

For a complete characterization of the structure of concrete specimens, ultrasonic tests were carried out after 2, 7, and 28 days of hardening (exposure) (Figure 9). According to the results of the ultrasonic study, it can be concluded that WMP and pozzolanic additives significantly affect the structure of concrete both in the early periods of hardening and during the subsequent curing of the samples. Compared to the UPV values after 24 h of curing, the UPV values after curing for 2 days in the P0, PA, PAS, PAMK, and PASMK composition samples increased by 19.75%, 20.7%, 19.0%, 22.7%, and 18.6%. But it was seen that in the composition with WMP fillers only, the UPV values were the lowest and varied by 301 m/s from other compositions. The WMP fillers, compared to traditional fillers aggregates, hindered the development of the structure. It is known that pozzolanic additives participate in the pozzolanic reaction and start reacting just after the release of the calcium hydroxide from the primary hydration of cement minerals and quickening of the structure densification [62,63,64,65].

After 7 days of hardening, the UPV growth rates slowed. The largest change in UPV was observed in the P0 and PA composition samples—7.3% and 5.7%, compared to hardening values after 2 days. When the pozzolanic additives of microsilica and metakaolin were used, the change in UPV was 5.0% and 5.1%, but when both metakaolin and microsilica additives were used together, the change in UPV was 4.7%, although the UPV values in this composition samples were the highest and reach 4252 m/s. After 28 days of hardening, the previously observed tendencies are confirmed—the composition PA with WMP fillers only showed the lowest UPV values, and the pozzolanic additives in compositions sped up the structure development. During the hardening periods, UPV values mostly increased in the PAMK composition samples by 28.7%, and at least by 22.7% in the PASMK composition samples. 

In the samples P0, PA, and PAS, the UPV values increased by 27.0%, 28.3%, and 24.8%. Despite the UPV growth trend, it can be concluded that the denser (by almost 1.9 times) samples of PA without pozzolanic additive showed significantly lower UPV values than the control P0 samples after 24 h, 2, 7, and 28 days of hardening. These results indicate incomplete contact between the WMP and the cement matrix. The use of pozzolanic additives, especially the combined use of microsilica and metakaolin additives, provided a less defective structure than in the PA samples. The pozzolanic additives, due to their high degree of dispersion, provided a denser cement matrix structure [54,55,56,66,67,68] and better contact between the metal particle and matrix.

#### 3.2.2. Microstructure

The microstructure of the samples (PA, PAS, PAMK, and PASMK) with WMP and pozzolanic additives were studied with the same magnifications (500 times). It is obvious that contact between the WMP and cement matrix in the PA sample was unsatisfactory (Figure 10a). The air gap size was about 10–15 µm. In the case of the metakaolin additive (Figure 10b), the contact between the WMP and the cement matrix was satisfactory; however, it can be seen that the matrix itself was somewhat more friable, which visibly reflects that the density and UPV of this sample are lower. When in the composition additive microsilica and both additives microsilica and metakaolin are used (Figure 10c,d), the adhesion between WMP and cement matrix is visibly improved, and the matrix seemed denser. The effects of microsilica on microstructure [32] are explained by the effect of microsilica fineness, as well as the effect of the pozzolanic reaction. These enhanced the amount of products leading to a pore refinement and further densification of the interfacial transition zone between the WMP and the cement matrix and strengthening of the bond between the WMP and the cement matrix.

#### 3.2.3. Compressive Strength Tests

The results of the compressive strength tests of the samples after 2 days of hardening are presented in (Figure 11). In comparison with the control WMP-free samples, the used WMP fillers, as well as in combination with the pozzolanic additives of microsilica and metakaolin reduced the compressive strength of the mortar sample by 18.5% and by 7.3–11.1% (in case of microsilica and metakaolin additives) after 2 days of hardening. However, the use of additives makes it possible to achieve the strength of 25 MPa, which is just 7.3% lower than the strength of specimens of control composition. It can be emphasized that the use of particles with an irregular shape and an uneven surface in composition results in higher compressive strength [3]. In our case, using the WMP fillers of a regular shape and smooth surface resulted in the lower strength of the tested samples.

In the research [15], it was proved that increasing the waste steel content in concrete from 0 to 50% by mass led to a decrease in the compressive strength of the sample by 20 times (from 50 MPa to 3 MPa). In our study, a high amount of WMP (45% of WMP by mass in composition) was used; however, the lowest compressive strength value is 20 MPa, which is satisfactory for 2 days hardened sample.

These compressive strength results confirm the UPV studies (Figure 9), showing that during the first 6–7 h of hydration, the structure of the PASMK and PAS specimens densified faster, which indicates the accelerated formation of crystal hydrate nuclei [62,69,70,71], such as calcium aluminate ferrite hydrate, calcium aluminate hydrate and calcium silicate hydrate. As pointed out in [37,72,73], the microsilica additive intensifies the cementitious material’s strength significantly because it maximizes the bond strength between the aggregate particles and the cement paste. Even the small additions of the microsilica (2–5%) affect the interfacial transition zone by producing a dense structure, with an increase in fracture toughness and microhardness.

In the case of the WMP filler, it was obvious that the metal increased the density of the sample, but at the same time, the metal filler, because of its regular shape and smooth surface, weakened the structure, as can be seen from the microstructure (Figure 10a) and compressive strength due to the formation of a more defective structure.

After 7 days of hardening, the same trends were observed—samples of the P0 and PASMK compositions differ from others in the highest compressive strength, which reaches 32.8 and 33.5 MPa. The fine pozzolanic particle is beneficial for the cementitious materials because they can fill the voids among the large cement particles and the primary CASH gel network of Portland cement, producing a dense cement paste matrix [72].

The strength of the specimens of the composition PA reached 26.0 MPa, composition PAS 29.8 MPa, and composition PAMK 28.0 MPa. The cement gel, which is composed of crystallized calcium aluminate ferrite hydrate, calcium aluminate hydrate, and calcium silicate hydrate, hardens over time to form a continuous binder matrix with a large surface area, and its constituents are responsible for paste strength gain [71]. As pointed out in [74], the pozzolanic reactions are very slow and usually initiate after one week from the inception of the hydration reaction.

After 28 days of hardening, the PA composition samples still have the lowest compressive strength values (36.0 MPa). However, the combined use of WMP and both additives make it possible to achieve the strength of 47.0 MPa, with microsilica and metakaolin additives used separately—44.0 and 43.0 MPa. PASMK compositions samples’ compressive strength is 22% higher than in composition PA. It can be supposed that the faster structure formation revealed in the UPV (Figure 9) research results indicates the accelerated formation of products of the pozzolanic reaction [62,69,70,71,74] and results in higher mechanical properties and denser structure than the common types of such materials. The results of compressive strength, standard deviation and coefficient of variation, and all calculations are presented in Table 5. The coefficient of the standard deviation of experimental results varies in the range 0.63–2.29 MPa; the coefficient of variation varies in the range 2.52–7.87%. These calculations of the standard deviation and coefficient of variation revealed the reliability of the obtained results.

In sum, we can conclude that there was an interrelation between the compressive strength values and the values of UPV for the tested samples: with an increase in the strength of the samples, the UPV values increased. However, the PA composition samples, because of unsatisfied contact between WMP particle and cement matrix (air spaces are clearly visible), possessed the lowest compressive strength and UPV values accordingly. However, for the density values and UPV values of the PA samples, such an interrelation for the following reason was not observed.

#### 3.2.4. Water Absorption Test

Water absorption is closely related to the porosity of the material [75] and the type and properties of the used waste material [76]. The water absorption test provides access to the open but not to the closed porosity, which remains undetermined together with the material’s total porosity. However, low water absorption reveals increased compressive strength [77]. Compounds of the mortar mixture highly influenced the water absorption results. The 28-day water absorption values of the control P0 samples were found as 5.2%; for the PA samples—8.36% (Figure 12). This result can be related to the higher porosity developed by WMP fillers in the PA composition sample (due to poor contact between WMP and the matrix), compared to common fillers. As is pointed out in the research [78], the metal fillers-based concrete had more water present in the concrete compared to traditional concrete and resulted in more voids and a more porous structure once the concrete set.

We can state that replacing the part of sand with WMP fillers in the composition significantly increased the water absorption of the PA samples. Because the fillers made up around 75% of the volume of the concrete, the particle form, gradation, and maximum size had a substantial impact on the concrete’s overall behaviour and qualities [37]. In the case of the PA sample, the higher water absorption values can be based on the fact that the air gaps between WMP fillers and the matrix exist.

The decreasing values of water absorption in the PAS, PAMK, and PASMK samples can be attributed to the pozzolanic activity of the microsilica and metakaolin additives [79]. It is observed from the microstructure tests that both the pozzolanic additives created a denser matrix, improved the contact between the WMP and cement matrix (Figure 10b–d) and reduced the water absorption of PAS and PAKM samples by up to 6.2% and 5.9%. The use of both additives together in the PASMK sample allowed for a further reduction in water absorption by up to 5.8%. In general, the use of pozzolanic materials in the composition with WMP caused a decrease in water absorption values. As it was concluded in [69], the effect of the pozzolanic reaction made the cementitious materials strong and durable. This is practically done by producing additional cement gel, reducing the pore extent, blocking its capillary and producing dense concrete, and increasing the adhesion between the cement matrix and fillers.

#### 3.2.5. Sorptivity Test

Figure 13 shows the capillary water absorption coefficients (C_w_) of samples, which refer to the absorption of water into the sample structure through its capillaries, indicating the pore structure and connectivity (capillary network) [60]. A significant increase in the capillary absorption coefficient was observed with the replacement of part of the sand fillers with WMP fillers. In the P0 samples, after 10 min of sorption, the capillary water absorption coefficient reached 1.03 g/m^2^·min, but in the PA sample—1.55 g/m^2^·min. The capillary rise of water increased, resulting in the increasing weight of concrete. However, despite the high density of samples, the capillary water absorption coefficient of PAS, PAMK, and PASMK samples were 1.14, 0.99, and 0.97 g/m^2^·min, respectively. According to [3,34], more than the cement fineness of the pozzolanic microsilica and metakaolin additives, it is the powder that promotes filling the voids and a dense microstructure, and improves resistance to permeability in capillary [80,81]. 

The results showed that the capillary water absorption coefficient gradually decreased for all composition samples with an increase in the time of sorption. The highest capillary water absorption coefficient after 1440 min was observed for PA composition samples—0.08 g/m^2^·min, the lowest—0.034 g/m^2^·min for P0 samples. Between the compositions with pozzolanic additives, the lowest capillary water absorption coefficient is observed for PAMK and PASMK samples—0.037 and 0.036 g/m^2^·min. This result clearly revealed the positive effect of both the pozzolanic additives of microsilica and metakaolin in the compositions with WMP fillers on the capillary water absorption values.

## 4. Conclusions

1. The influence of WMP fillers and the microsilica and metakaolin pozzolanic additives on the hydration process of PC were investigated. The partial replacement of sand by WMP fillers in the mortar mixes by 32.0% reduces the wetting heat release rate. The addition of microsilica and metakaolin additives reduces the wetting heat release rate by 12.0% and 11.0%. When microsilica and metakaolin additives are used together, the wetting heat release rate decreases by 6.0%.

The WMP particles prolonged the PC hydration course and increased the heat release rate: the maximal heat release rate for mortar mix with partially substituted sand to WMP is 26% higher than in the control mix without the WMP particles. Compared to the control mortar mix, the microsilica and metakaolin additives used in the mix with WMP fillers increased the maximum heat release rate value and accelerated hydration by 37.5% and 27.5%. When the microsilica and metakaolin additives were used together in the mix, the maximum heat release rate value increased up to 46.0%.

The results show that the microsilica additive and the microsilica and metakaolin additives mostly stimulate the hydration process of PC mortar. After 24 h of hydration, the lowest total heat value was reached in the control mix, but partially substitution of sand by WMP fillers increased the total heat value by 17.0%. The addition of microsilica, metakaolin, and both microsilica and metakaolin additives increase total heat value by 30.1%, 34.2%, and 37.0%. It can be concluded that WMP fillers increased the total heat value in the mixes, especially when microsilica and metakaolin additives are used together. These findings are important because of the use of metal fillers in massive structures.

2. WMP fillers and pozzolanic additives retarded the early phase of the PC hydration process. After 24 h of hydration, the UPV value in the sample with partially substituted sand to WMP was 4.0% lower than in the control sample. When WMP fillers collect the hydration heat, the structure development runs conventionally, and after 24 h of hydration, the structure densification development reached almost the same level as in the control sample. The addition of a mix of microsilica and metakaolin and separately used microsilica and metakaolin additives increased the UPV values up to 16%. When WMP is used in combination with microsilica and metakaolin additives, the additives provided a less defective structure of the samples, and the structure development and densifying of the sample was faster.

3. A partial substitution of sand with WMP fillers in the mortar mix increased the density of the sample from 2180 to 4170 kg/m^3^. The addition of the pozzolanic additive slightly reduced the sample density from 3985 to 4130 kg/m^3^. The values of UPV and density of samples, where WMP fillers partially substitute sand, did not correlate with each other; this is associated with incomplete contact between the WMP fillers and the cement matrix, which occurred during the early phase of the hydration of cement. The UPV in denser samples with WMP filler was lower than in control concrete samples.

4. Due to the smooth surface of WMP, the partial substitution of sand with WMP in the mix up to 18.5%, 20.8%, and 14.2%, reduced the compressive strength of the concrete samples after 2, 7, and 28 days of hardening in comparison with the control samples. These results are also confirmed by the UPV results. When used together with WMP fillers, the microsilica and metakaolin additives and both additives together increase the compressive strength of concrete samples after 2 days of hardening by 12.0%, 8.3%, and 12.1%, after 7 days of hardening by 10.6%, 7.1%, and 16.13%, after 28 days of hardening by 16.2% both and 21.8% in comparison to the samples where the sand was partially substituted to WMP. The combined use of both the microsilica and metakaolin pozzolanic additives allowed for the air gap to be avoided between the WMP and the cement matrix. The pozzolanic additives promoted better contact between the cement matrix and the WMP and allowed for a strength of 46.50 MPa to be achieved after 28 days of hardening, which is 8.7% higher than the strength of the control samples. The combined use of pozzolanic additives improved the structure development.

5. The microstructure, compressive strength, and UPV tests were confirmed by water absorption and capillary water absorption coefficient research. It was observed that both the water absorption and capillary-adsorbed water in the samples increased when the sand was partially substituted by WMP fillers in the samples. The samples where the sand was partially substituted by WMP fillers possessed poor structure by a 36.5% higher water absorption value and a 1.5 to 2 times higher capillary water absorption coefficient during the first 90 min than in the control sample. The combined use of the microsilica and metakaolin pozzolanic additives allows for the reduction of water absorption by 33% and the capillary water absorption coefficient up to 56.6% during the first 90 min compared to the WMP-containing samples without pozzolanic additives.

In sum, we can conclude that pozzolanic additives can solve difficulties in the application of local waste-metal particle fillers in heavyweight concrete. The developed heavyweight concrete is suitable for being used in energy facilities, hydrotechnical foundations, dams and load balancing.

## Figures and Tables

**Figure 1 materials-15-02925-f001:**
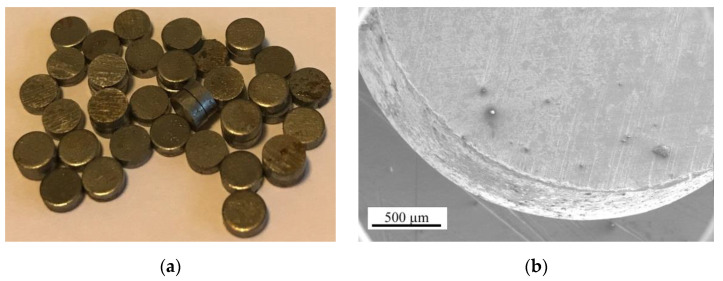
Image of the waste-metal particle: (**a**) overall view, (**b**) SEM separate particle.

**Figure 2 materials-15-02925-f002:**
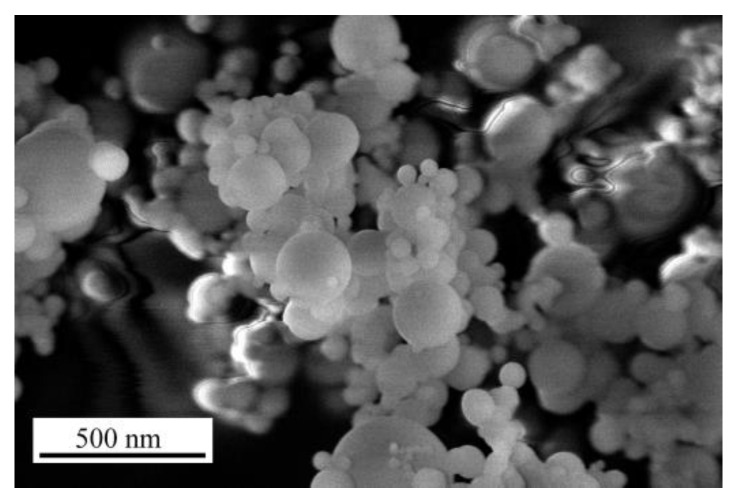
SEM image of the microsilica particles in µm.

**Figure 3 materials-15-02925-f003:**
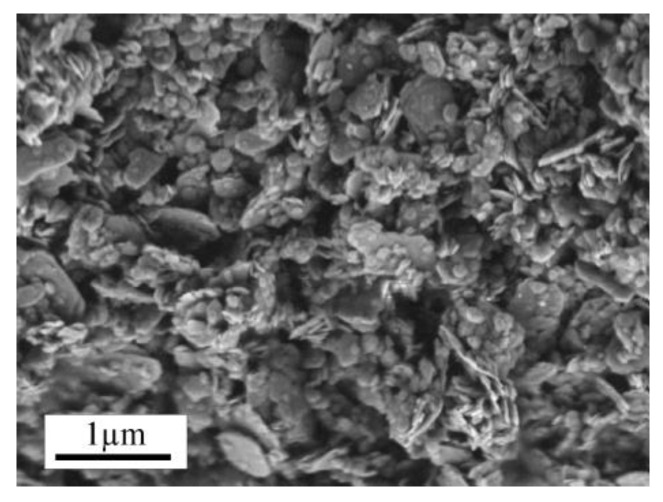
SEM image of the metakaolin particles.

**Figure 4 materials-15-02925-f004:**
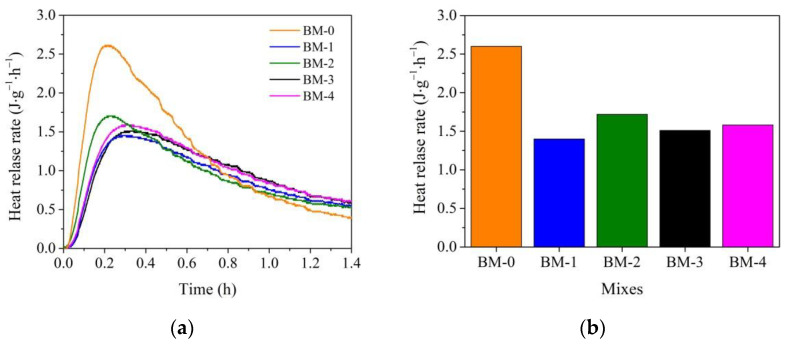
Wetting heat release rate for BM-0–BM-4 mixes: (**a**) curves, (**b**) maximum values during first 20 min.

**Figure 5 materials-15-02925-f005:**
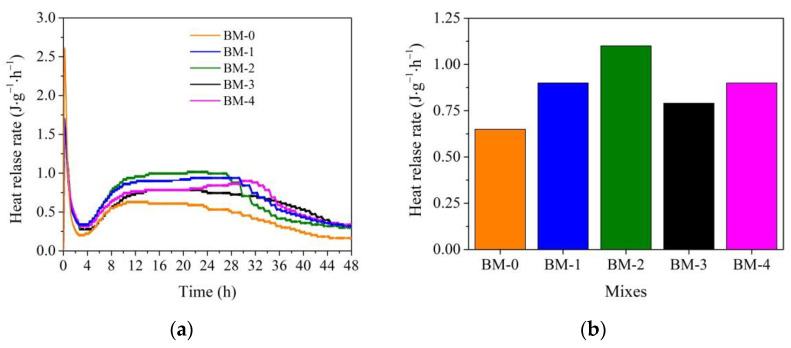
The heat release rate for BM-0–BM-4 mixes: (**a**) curves; (**b**) maximum values.

**Figure 6 materials-15-02925-f006:**
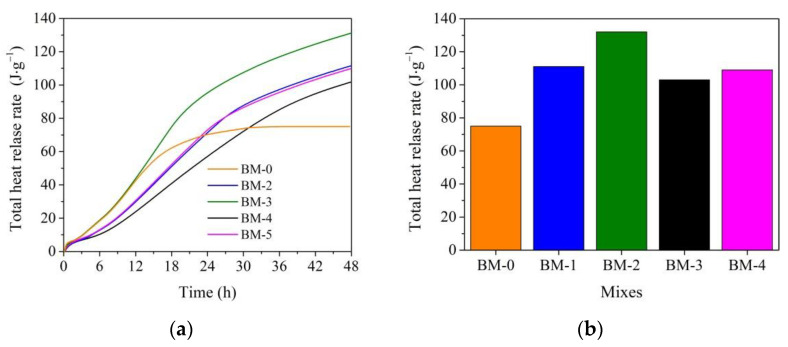
The total heat release rate for BM-0–BM-4 mixes: (**a**) curves; (**b**) maximum values.

**Figure 7 materials-15-02925-f007:**
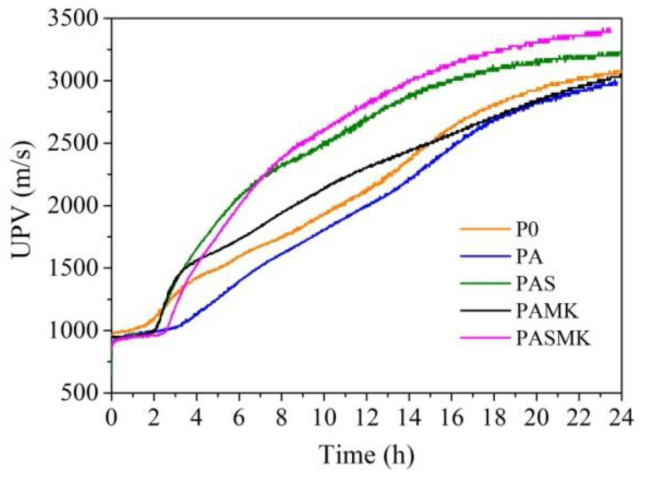
The variation of UPV in fresh mortar mixes P0–PASMK samples during 24 h.

**Figure 8 materials-15-02925-f008:**
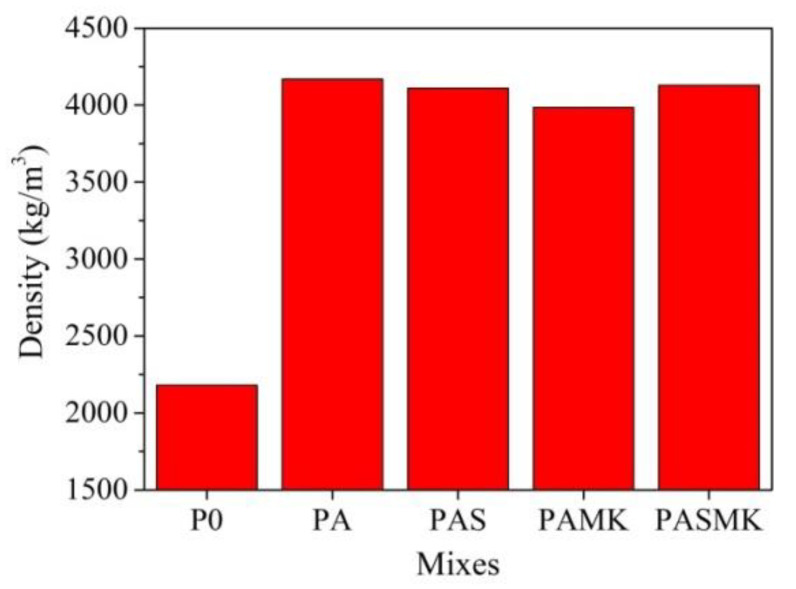
The density of P0—PASMK samples.

**Figure 9 materials-15-02925-f009:**
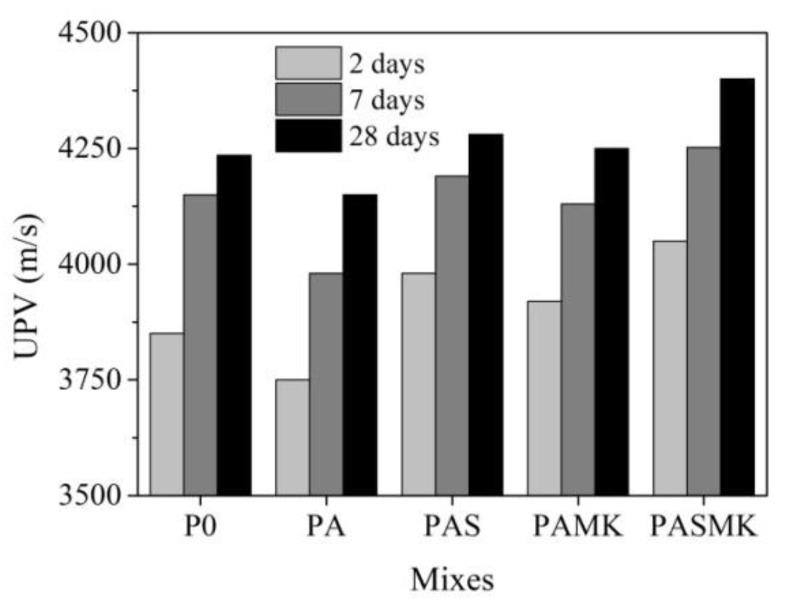
The variation of UPV in samples P0–PASMK samples for 28 days.

**Figure 10 materials-15-02925-f010:**
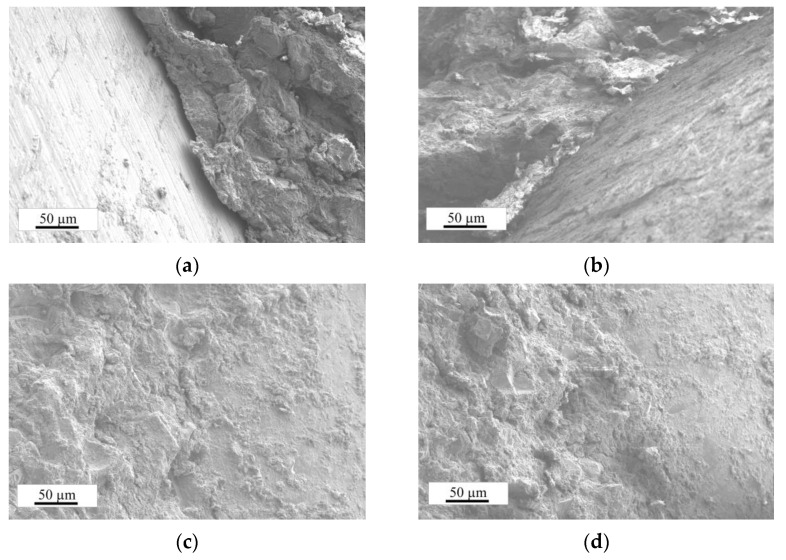
SEM image of the composition sample: (**a**) PA; (**b**) PAMK; (**c**) PAS; (**d**) PASMK.

**Figure 11 materials-15-02925-f011:**
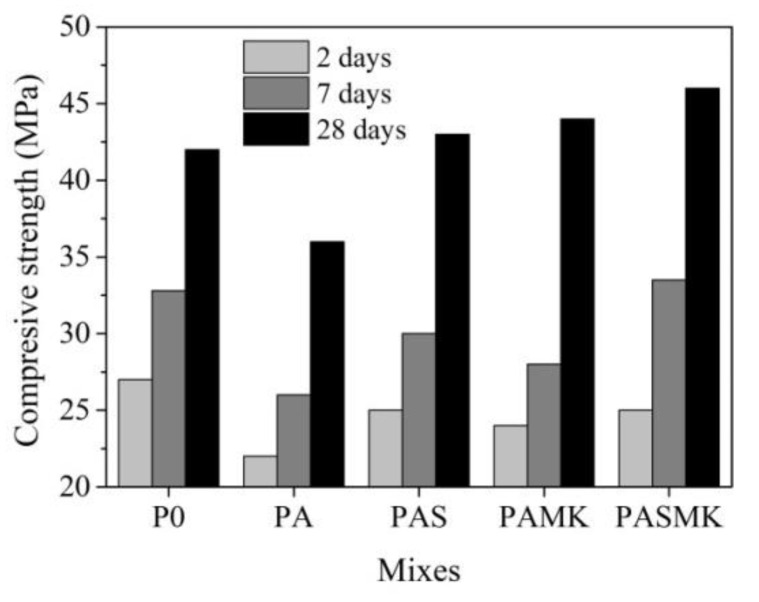
The compressive strength of P0–PASMK samples after 2, 7, and 28 days.

**Figure 12 materials-15-02925-f012:**
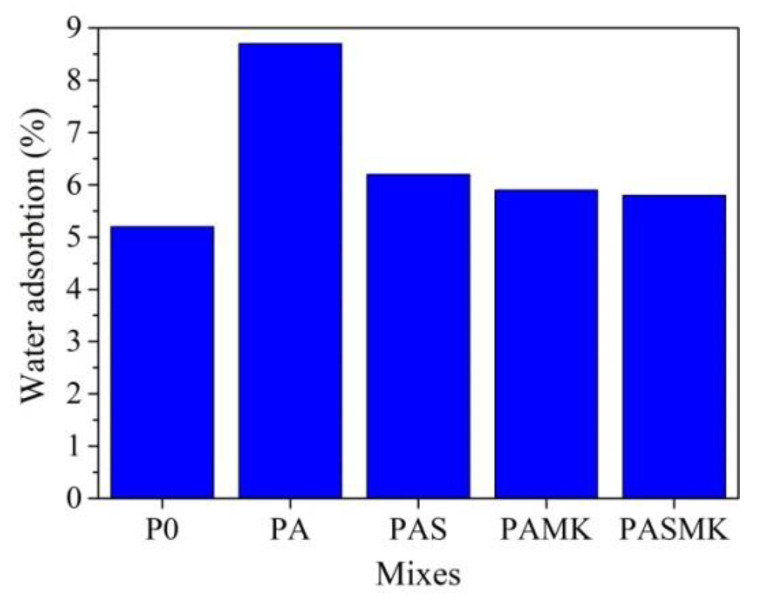
Variation in the water absorption by samples with different fillers and pozzolanic additives.

**Figure 13 materials-15-02925-f013:**
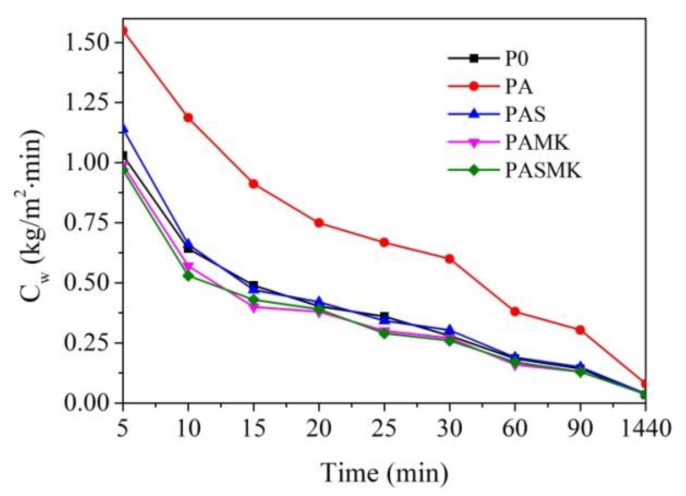
Capillary water absorption coefficient (C_w_) of specimens P0–PASMK.

**Table 1 materials-15-02925-t001:** Chemical composition of PC, microsilica, and metakaolin.

Components	Chemical Composition (%)							
	SiO_2_	Al_2_O_3_	Fe_2_O_3_	CaO	MgO	K_2_O + Na_2_O	C	LOI
Portland cement	20.8	6.12	3.37	63.5	–	1.03	–	0.30
Microsilica	98.0	0.30	0.05	0.30	0.10	0.30	0.40	0.60
Metakaolin	52.1	45.6	0.50	0.20	0.20	0.30	–	1.10

**Table 2 materials-15-02925-t002:** Physical properties of sand.

Aggregate	Fraction	Particle Density, kg/m^3^	Water Adsorption, %	Bulk Density, kg/m^3^
Sand	0/4	2345	0.53	1595

**Table 3 materials-15-02925-t003:** Compositions of mixes with microsilica and metakaolin additives and WMP particles for calorimetric investigations (components mass by weight, g).

Batch	Components					
	PC	Sand Fraction 0/1	Microsilica	Metakaolin	WMP	W/S
BM-0	35	65	–	–	–	0.45
BM-1	35	20	–	–	45	0.45
BM-2	35	–	20	–	45	0.45
BM-3	35	–	–	20	45	0.45
BM-4	35	–	10	10	45	0.45

**Table 4 materials-15-02925-t004:** Compositions of mortar forming mixtures (mass %).

Batch	Components					
	PC	Sand Fraction 0/4	Microsilica	Metakaolin	WMP	W/C
P0	20	80	–	–	–	0.55
PA	20	35	–	–	45	0.55
PAS	20	25	10	–	45	0.58
PAMK	20	25	–	10	45	0.58
PASMK	20	25	5	5	45	0.58

**Table 5 materials-15-02925-t005:** Average values of the compressive strength of the studied concrete.

Batch	Testing Age in Days	Compressive Strength (MPa)	SD (MPa)	COV (%)
P0	2	27.3	1.42	5.22
	7	32.5	2.29	7.06
	28	41.5	1.26	3.04
PA	2	21.8	1.29	5.90
	7	25.6	1.79	7.02
	28	35.7	1.46	4.08
PAS	2	25.1	1.18	4.71
	7	29.9	1.89	6.34
	28	42.8	1.67	3.89
PAMK	2	23.9	0.93	3.90
	7	28.3	2.22	7.87
	28	44.0	1.68	3.81
PASMK	2	25.1	0.63	2.52
	7	33.4	1.93	5.78
	28	46.6	1.87	4.03

## Data Availability

Not applicable.

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
