# Peer review of "Study of the Course of Cement Hydration in the Presence of Waste Metal Particles and Pozzolanic Additives"

_materials, 2022, doi:10.3390/ma15082925_

Round 1

Reviewer 1 Report

The article presents the study of cement hydration in the presence of waste metal particles and pozzolanic additive. The article does not present the basic properties of concrete components in a tabular form. The work contains a lot of inaccuracies and erroneous markings. Some wording requires editing.

The article is not suitable for publication in Materials as it stands.

General remarks

  1. As for me, in a separate subsection it should be clearly defined what is the purpose and scope of the research.
  2. The research description should be properly formulated in the introduction. In many cases, inappropriate technical terms were used.
  3. Information on the composition of the individual components should be presented in the form of a 137-157 lines table.
  4. The meaning of adding CMM in terms of the compressive strength of the concrete should be clearly explained. The authors use the addition of microsilica and metakaolin to compensate for the decrease in strength.
  5. What quantities of samples have been tested in the test methods presented? What were the statistical parameters (SD and COV)?
  6. On what samples were the strength tests of the samples carried out? How many samples were there? What was the standard deviation and the COV?
  7. Moreover, in many cases the syntax of sentences needs to be improved.

The article is not well written.

Specific remarks

  1. 26 line: “... the compressive strength of WMP containing mortar by 22%.” In relation to what value?
  2. 34 line: Better: “components” no "composition".
  3. 40 line: Use kg/m3 units consistently.
  4. 44 line: Please list what types of fillers.
  5. 48 line: Please put in the density table and then discuss briefly.
  6. 56 line: Give specific examples.
  7. 57 line: What does "high wear resistance" mean?
  8. 62 line: What does "incorrect handling" mean?
  9. 64 line: To be added: "Segregation of ingredients…. ".
  10. 71 line: Please specify which composite?
  11. 73 line: Should be: metallic minerals, not metals.
  12. 74 line: What specific properties have improved?
  13. 80 line: Please specify which composite?
  14. 84 line: Should read: "increase in the proportion of barite"
  15. 101 line: Please specify the specific strength degradation?
  16. 105 line: A density of steel 7.8 kg/m3 - unbelievable! What was the percentage of "steel shot"?
  17. 106 line: How big was the share of the "steel shot" in the concrete?
  18. 121 line: Should be: "in concrete".
  19. 123 line: Should be: "as concrete modifying additives".
  20. 130 line: Should be: Materials and research methods.
  21. 141, 156 lines: Please explain in the paper what the term "loi" means.
  22. 141 line: The specific surface should be in the same units as before.
  23. 149 line: Should be: “calcination loss at 975 °C

31.175, 179, 192, 193 etc. lines: "microsilica" or "microsilika"?

  1. 235 line: Explain which tests were performed and by whom.
  2. 237 line: Please correct the syntax of the sentence.
  3. 270 line: Which drawings are concerned with this note?
  4. 290 line: Better "Ultrasound Propagation Velocity Method"
  5. 343 line: Should be: “Properties of hardened samples”.
  6. 353 line: Should be: lowercase "w/c".
  7. 457 line: Subsection numbering should be changed.
  8. 491 line: Subsection numbering should be changed.
  9. 514 line: Should be: "Conclusions".

I recommend an in-depth review of the manuscript, including comments, to make it an article suitable for publication in the Materials.

Reviewer 2 Report

The article is about cement hydration course in the presence of waste metal particle and pozzolanic additive. However, some issues must to be addressed:

  1. Abstract: Please start by expressing the aim of this paper, followed by the rest of the information. Also, please define or try to avoid using abbreviations in the abstract. Typically, the abstract should provide a broad overview of the entire project, summarize the results, and present the implications of the research or what it adds to its field.
  2. Please avoid bulk citation like [13-16], [27-31] and rephrase the introduction section together with improvement of reference section.
  3. The results are merely presented, not properly discussed. Please add explanations for the observed changes. Please give an extended discussion on the obtained results and correlate your findings with previous literature studies and prospective applications.
  4. More analysis and interpretation of the results should be added for a clearer understanding of observed experimental phenomena.
  5. The authors must to provide some details about importance of the research and their applicability.
  6. Please introduce the conclusion section in order to highlight the results obtained.
  7. General check-up and correction of the English language is suggested. There are still some minor typos and grammatical errors.

The author needs to address the abovementioned points for the betterment of the manuscript.

Reviewer 3 Report

This manuscript is about the study of cement hydration course in the presence of waste metal particle and pozzolanic additive

This manuscript needs some improvements.

Abstract:

Add something about the mechanical performance/ results of the research. 

Add something about the benefits results of the research. 

Introduction

-Well written.

Materials and Methods:

-A comprehensive research framework missing- to follow the research is steps are missing. Add framework-flowchart and write this section in stepwise pattern.

- mechanical and durability performance missing…. Compressive only available, flexural strength missing? (Major Issue)

Results:

In this section add comparative analysis of your results with previously published papers (4-5 reference results) (i.e., comparison with compressive, flexural performance etc.)..

Please compare the results of this research within existing published research with existing materials. Line or bar charts can be added. [Please add]

Discussion:

This section must contain implications for research, practice and/or Field: Does the paper identify clearly any implications for research, practice and/or society? Does the paper bridge the gap between theory and practice? How can the research be used in practice (economic and commercial impact), to influence technical policy, in research (contributing to the body of knowledge)? Add something for field professionals. [Please add]

Limitations of the study:

Please add as heading about the limitations of the study.

Round 2

Reviewer 2 Report

The article is suitable for publication.

Reviewer 3 Report

This manuscript can be accepted in current form.